# Quantitatively Recognizing Stimuli Intensity of Primary Taste Based on Surface Electromyography

**DOI:** 10.3390/s21216965

**Published:** 2021-10-20

**Authors:** Hengyang Wang, Dongcheng Lu, Li Liu, Han Gao, Rumeng Wu, Yueling Zhou, Qing Ai, You Wang, Guang Li

**Affiliations:** State Key Laboratory of Industrial Control Technology, College of Control Science and Engineering, Zhejiang University, Hangzhou 310000, China; 11432014@zju.edu.cn (H.W.); 3180103662@zju.edu.cn (D.L.); liliu@zju.edu.cn (L.L.); gao_han@zju.edu.cn (H.G.); rumeng@zju.edu.cn (R.W.); 3180102168@zju.edu.cn (Y.Z.); aiqing@zju.edu.cn (Q.A.); king_wy@zju.edu.cn (Y.W.)

**Keywords:** primary taste, taste stimuli intensity, pattern recognition, Support Vector Machine (SVM), brain-computer interface (BCI), surface electromyography (sEMG)

## Abstract

A novel approach to quantitatively recognize the intensity of primary taste stimuli was explored based on surface electromyography (sEMG). We captured sEMG samples under stimuli of primary taste with different intensities and quantitatively recognized preprocessed samples with Support Vector Machine (SVM). The feasibility of quantitatively recognizing the intensity of Sour, Bitter, and Salty was verified. The sEMG signals were acquired under the stimuli of citric acid (aq), sucrose (aq), magnesium chloride (aq), sodium chloride (aq), and sodium glutamate (aq) with different concentrations, for five types of primary tastes: Sour, Sweet, Bitter, Salty, and Umami, whose order was fixed in this article. The acquired signals were processed with a method called Quadratic Variation Reduction to remove baseline wandering, and an adaptive notch to remove power frequency interference. After extracting 330 features for each sample, an SVM regressor with five-fold cross-validation was performed and the model reached *R*2 scores of 0.7277, 0.1963, 0.7450, 0.7642, and 0.5055 for five types of primary tastes, respectively, which manifested the feasibilities of the quantitative recognitions of Sour, Bitter, and Salty. To explore the facial responses to taste stimuli, we summarized and compared the muscle activities under stimuli of different taste types and taste intensities. To further simplify the model, we explored the impact of feature dimensionalities and optimized the feature combination for each taste in a channel-wise manner, and the feature dimensionality was reduced from 330 to 210, 120, 210, 260, 170 for five types of primary tastes, respectively. Lastly, we analyzed the model performance on multiple subjects and the relation between the model’s performance and the number of experiment subjects. This study can provide references for further research and applications on taste stimuli recognition with sEMG.

## 1. Introduction

In human brains, the taste stimuli from the outside world generate taste sensations, which play a crucial role by motivating and regulating human feeding processes [1]. Many studies have focused on the recognition of taste based on the recognition of the chemical composition of the stimuli source by sensors. The electronic tongue is a type of analytical instrument that comprises an array of low-selective, nonspecific, chemical sensors and a data processing method [2], which have been applied to many fields, such as the food industry [3], water analysis [4], pharmaceutical industry [5], and clinical diagnosis [6]. Biomimetic sensors and biosensors have also been applied for taste recognition [7]. The biomimetic sensor is a type of sensor with sensing elements based on biomimetic materials. For instance, an artificial lipid polymer membrane has been applied to taste recognition of Sweet [8]. The biosensor is a type of sensor with elements based on biological principles. Sensors based on taste receptor cells can distinguish Sweet and Bitter [9], and sensors based on enzymes can recognize the chemical components of wine [10].

However, when we try to compare the taste stimuli of two different tastants, it is difficult to do so, even if we know their chemical compositions with the help of traditional sensors. Moreover, with the rapid development of virtual reality, many studies have focused on virtual taste [11], a technology to provide a taste sensation experience without real tastants [12,13,14], in which traditional sensors are scarcely applied. However, the human brain can generate similar responses to similar stimuli. Therefore, a brain-computer interface (BCI), a system receiving, analyzing, and transferring physiological signals into the real world could solve the problem [15,16]. By recognizing the patterns of physiological signals, BCI could give a description of taste stimuli, whether it is a real tastant or a virtual taste device. BCI has shown its potential in taste sensation recognition, utilizing physiological signals such as functional Magnetic Resonance Imaging (fMRI) [17] and Magnetoencephalography (MEG) [18]. However, these physiological signals were limited by the requirements for the site and large equipment. Among many physiological signals in BCI applications, electroencephalogram (EEG) is popular for its non-invasiveness, information integrity, and device portability [19,20,21] and has been applied for the recognition of taste sensations [22,23,24,25]. However, EEG also has shortcomings [26], such as irrelevant channels, overwhelming information, and the complexity of signal acquisition and processing, which could be solved by surface electromyography (sEMG), benefitting by pasting electrodes specifically on the related muscles and avoiding the influence of skull and hair.

sEMG monitors the electrical activity of motor units, a constitutional unit combining motor neurons and muscle fibers, through which it non-invasively records the activity of muscle [27]. The motor units are driven by the central nervous system. Therefore, sEMG also has information related to the activity of the central nervous system and could be a favorable supplement or alternative to EEG. sEMG has proven its effectiveness in many fields, such as speech recognition, gesture recognition, etc. [28,29,30,31]. There are some studies on sEMG during the eating process. Sato analyzed the effect of taste on sEMG during consumption [32] and Miura analyzed the effect of taste, carbonation, and temperature on sEMG during swallowing [33]. Manda analyzed the effect of food properties and chewing conditions on sEMG during chewing [34]. However, the process of swallowing or chewing is different from stable stimuli of taste without voluntary muscle movements. Under taste stimuli, a human would show facial expressions responding to the taste experience and it was shown that sEMG could record and recognize the pattern of facial expressions [35,36]. Moreover, some studies have confirmed that with the help of facial expressions, the sEMG could record the facial muscle activities responding to the taste stimuli [37,38,39]. Hu analyzed the activity of levator labii superioris under the stimuli of five different drinks and verified the taste stimuli providing negative hedonic value could induce the activity of levator labii superioris [37]. Horio evaluated facial expression patterns induced by eight tastants and verified that the activities of facial and chewing muscles (such as corrugator, orbicularis oculi, masseter, risorius, orbicularis oris, depressor anguli oris, and digastricus) under stimuli of different tastes had statistically significant differences [38]. Armstrong investigated the facial response of children to four different tastants and verified that activities of zygomaticus and levator labii superioris were related to taste stimuli and that this response also exists for children [39]. These studies were similar, except for the selection of muscle and tastant, and verified that the sEMG under different taste stimuli showed statistically significant differences. However, the study on recognizing different taste stimuli is still under investigation, which is the problem we are concerned about in this study. We aimed at utilizing the sEMG response and making a recognition model for taste stimuli so that we could describe the taste stimuli with patterns of physiological signals in a standard manner. The generally accepted primary tastes include tastes of Sour, Sweet, Bitter, Salty, and Umami [40]. Most complex taste stimuli could be considered as superpositions of primary taste stimuli [1]; therefore, a quantitative recognition of primary taste stimuli intensities could be a foundation for establishing this standard description of taste stimuli.

In this study, a quantitative approach to recognize the stimuli intensity of primary taste based on sEMG was proposed, as shown in Figure 1. Experiments were designed and carried out to acquire the sEMG signals of five taste types with six taste intensities each. Five types of tastant solutions were used: citric acid, sucrose, magnesium chloride, sodium chloride, and sodium glutamate [14,35]. For each type of solution, six different concentrations (including a concentration of zero) were used. After the acquisition, signals were augmented into samples. For each sample, a method called Quadratic Variation Reduction (QVR) was applied to remove baseline wandering [41] and an adaptive notch was applied to remove power frequency interference. After preprocessing, a feature set was extracted from each sample. Support Vector Machine (SVM) is a concise, classical, and popular pattern recognition method and Radial Basis Function (RBF) is a mainstream kernel function known for its good performance on non-linear problems [42,43]. SVM with a kernel function of RBF was performed on datasets with five-fold cross-validation, taking *R*2 scores as the main metric. To better describe the response of taste stimulus, we analyzed quantification results and the facial response with different taste stimuli. Moreover, to simplify the model, more efficient feature combinations were explored. Lastly, we analyzed the model’s performance on multiple subjects.

## 2. Material and Methods

### 2.1. Data Acquisition

Five subjects, three males and two females, participated in this study and provided consent. The age of the subjects ranged from 21 to 25, with an average age of 23.6. None of the subjects declare any medical history of taste disorders in the subject selection process and no diseases, medications, or other conditions associated with taste disorders were reported during the experimental period. The research was approved by the Ethics of Human and Animal Protection Committee of Zhejiang University.

With a six-channel sEMG data recording system, the sEMG data was directly captured from facial skin via standard, non-invasive Ag/AgCl electrodes, with a sampling frequency of 1000 Hz. The shape of the electrode was a circle with a diameter of 1.5 cm, and the diameter of the adhesive ring was 3.6 cm. Concretely, electrodes were arranged on facial and chewing muscles related to taste stimuli, including masseters, depressor anguli oris, levator labii superioris, risorius, and procerus, referring to the study of Horio and Hu [37,38]. The electrode distribution is shown in Figure 2. Electrode pairs 1 and 2 were differential electrode pairs while the other 4 electrodes were single electrodes. For a pair of differential electrodes, the adhesive rings’ edges of the two electrodes were tangent, which means the center-to-center distance of these two electrodes was 3.6 cm. Electrodes for channels 1 and 6 were arranged on masseters. The connection line between channel 1 electrodes’ centers was perpendicular to the fiber’s direction of the superficial masseter. Electrodes for channel 2 were arranged on depressor anguli oris and electrodes for channels 3, 4, and 5 were arranged on levator labii superioris, risorius, and procerus, respectively. The bias electrode is an electrode that inverts and amplifies the average common-mode signal back into the subject to significantly cancel common-mode interference and the reference electrode provides a reference potential for monopolar measurement. The bias electrode and reference electrode were arranged on the left and right mastoid, respectively. For all electrodes or electrode pairs, the electrode indexes, electrode types, and electrode positions (including the facial sides and the muscles) are shown in Table 1. The faces of the subjects were different in terms of shape and size, so the distances listed in the table should be fine-tuned according to the actual muscle positions.

To prevent the influence of food on taste sensation function of the subject from, the subject were asked not to eat an hour before the experiment; moreover, they were asked not to consume any irritating food twenty-four hours before the experiment. Each experiment contained two sessions and each session contained five primary tastes with six trials corresponding to six taste intensities for each primary taste. To provide taste experiences closely mimicking those of daily life, we utilized solutions of seasonings and food additives to provide primary taste, including citric acid (Youbaojia, anhydrous citric acid ratio is not less than 99.5%), sucrose (Taigu, sucrose ratio is not less than 99.6%), magnesium chloride (Tanggui, magnesium chloride hexahydrate ratio is not less than 98.1%), sodium chloride (Zhongyan, sodium chloride ratio is not less than 98.5%), and sodium glutamate (Xihu, sodium glutamate ratio is not less than 99.9%). The tastants providing different taste types and intensities are shown in Table 2.

Each experiment was carried out with the following steps.

(1)An experiment assistant checked the sEMG acquisition system.(2)The subject was asked to clean their face and the assistant checked that the subject was conscious and reported no diseases, medications, or other conditions associated with taste disorders.(3)The subject was asked to relax, and the assistant pasted electrodes on the face of the subject according to Figure 2.(4)The subject was asked to close their eyes. Then the assistant added 2 mL of the tastant solution to a new 35 mm diameter petri dish and soaked a piece of filter paper in the solution for 1 min.(5)The assistant put the filter paper on the tip of the subject’s tongue. The size and position of the filter paper are shown in Figure 3. The subject closed their mouth, keeping their tongue still. When the subject stopped moving their tongue or mouth and the signal baseline tended to be stable, the assistant started the device to get sEMG of the bitter taste for 12 s. During the 12 s, the subject was asked not to deliberately control their facial expression or to think about anything but the taste experience.(6)The subject gargled five times, taking 20 mL purified water each time.(7)The subject recorded the taste intensity score for the taste in step 5.(8)Steps 4 to 7 were repeated five times with the other five solutions of different concentrations.(9)The subject summarized scores and gave final scores for 6 types of solutions with different concentrations. The score was the evaluation of the taste intensity. If the subject considered the solution in a trial to be deionized water, the score should be 0 and if the subject considered that the solution in a trial had the strongest intensity, the score should be 10.(10)The subject took a break for two minutes.(11)Steps 4 to 10 were repeated four times with the other 4 solution serials of different primary tastes.(12)If there was another session, the subject took a break for 10 min. Then, steps 4 to 11 were repeated.

The order of different tastes and the order of different taste intensities for each taste was decided randomly. The duration for resting was decided by the time that the subject needed to fully recover from the last stimuli. For five subjects, 30 experiments were carried out, comprising 60 sessions.

### 2.2. Preprocessing

To augment samples, signals captured by the system were cut into samples by a sliding window. The length of a window was 1 s, and the step was 0.25 s. For each subject, during the earlier experiments, the assistant was asked to confirm that the subject’s facial expressions were continuous and stable until the stimuli ended. Meanwhile, the subject was also asked to confirm that both the taste of the paper and the saliva in their mouth were continuous and stable. As reported by the subjects and assistants, during the whole signal-capturing period (12 s), the filter paper provided continuous and stable stimuli on the tongue. Therefore, for each sample cut from the signal, the sEMG segment could reflect the sEMG character under corresponding stimuli.

The baseline wander, a global offset of the waveform, is usually caused by the apparent movement of facial muscles and zero drift of the device. To get rid of it, a QVR method was applied [41]. Assuming $\tilde{z}$ is the original signal and *z* is the signal after QVR, the mathematic relation between them is the following equation:(1)$$z=\left[I-{\left(I+\lambda D^{T}D\right)}^{-1}\right]\tilde{z}$$
where *λ* is a constant value (in this case λ=2), *I* is an identity matrix, and *D* is a $\left(n-1\right)\times n$ matrix:(2)$$D=\left[\begin{array}{ccccc} 1 & -1 & 0 & \cdots & 0\\ 0 & 1 & -1 & \ddots & \vdots \\ \vdots & \ddots & \ddots & \ddots & 0\\ 0 & \cdots & 0 & 1 & -1 \end{array}\right]$$
where *n* is the length of $\tilde{z}$ (in this case n=1000). In Equation (1), $I+\lambda D^{T}D$ is a symmetric positive-definite tridiagonal matrix, which can be solved efficiently.

The power frequency interference, a superposition of 50 Hz sine wave and its harmonics, was removed with an adaptive notch filter. The filter could modify a signal with an abnormally high amplitude at 50 Hz and its integer multiples without destroying most normal signals. For several samples, the power frequency interference was so heavy that we could only get a badly distorted signal after the notch filter. We had to remove these samples to ensure the quality of the data set.

An example of removing baseline wandering and power frequency interference is shown in Figure 4.

### 2.3. Feature Extraction

For each channel, the frequency range of the spectrum (from 0 to 500 Hz), was divided into 50 intervals and for each interval, the difference in frequency of the left and right ends was 10 Hz. For each interval, the integral value of the amplitudes of the spectrum was calculated as a feature. To further reflect energy distribution on different frequencies, four other frequency-domain features were also extracted, including frequency centroid (FC), root mean square frequency (RMSF), and root var frequency (RVF).

In addition, two time-domain features were also taken into consideration, including root mean square (RMS), and mean absolute value (MAV).

For each channel, 55 features were calculated according to the following equations:(3)$$F\left[n\right]=\sum_{i=n\times 10}^{\left(n+1\right)\times 10-1}f\left(i\right),\;when\;1\leq n\leq 50\;$$
(4)$$F\left[51\right]=\frac{\sum_{i=1}^{L/2}f\left(i\right)\times \left(i\times f_{s}/L\right)}{\sum_{i=1}^{L/2}f\left(i\right)}\;$$
(5)$$F\left[52\right]=\sqrt{\frac{\sum_{i=1}^{L/2}f\left(i\right)\times {\left(i\times f_{s}/L\right)}^{2}}{\sum_{i=1}^{L/2}f\left(i\right)}}\;$$
(6)$$F\left[52\right]=\sqrt{\frac{\sum_{i=1}^{L/2}f\left(i\right)\times {\left(i\times f_{s}/L\right)}^{2}}{\sum_{i=1}^{L/2}f\left(i\right)}}\;$$
(7)$$F\left[52\right]=\sqrt{\frac{\sum_{i=1}^{L/2}f\left(i\right)\times {\left(i\times f_{s}/L\right)}^{2}}{\sum_{i=1}^{L/2}f\left(i\right)}}\;$$
(8)$$F\left[55\right]=\frac{\sum_{i=1}^{L}\left|x_{i}\right|}{L}\;$$
where *F[n]* denotes the *n*th feature, *f(i)* is the spectrum amplitude of the *i*th point under the frequency domain, *L* is the length of a sample (in this case, L=1000), *f_s_* is the sampling frequency (in this case $f_{s}=1000$), and *x_i_* is the value of the *i*th data point of the signal under the time domain.

Since we got six channels and 55 features for each channel, the feature set we extracted contained 330 features.

### 2.4. Regression

SVM is a popular machine learning algorithm used for pattern recognizing tasks, including classifications and regressions. As a popular kernel function, rbf is capable of dealing with nonlinear problems. All SVM models in this study used rbf as the kernel function. For each primary taste type, we performed SVM on a dataset containing all samples of subject 1 to manifest the feasibility of the recognition task. Three types of labels, including a strength label (a label decreasing exponentially with the decrease of solution concentration), relative concentration (a label decreasing linearly with the decrease of solution concentration), and scale score (a label given by subjects during experiment according to their judgment of taste intensity) were attempted and the best one among them, the strength label, was fixed in the following steps of the study. Then, we analyzed the differences of facial expressions under stimuli of different taste intensities of the same primary taste type. Finally, to analyze the impact of subject diversity, we constructed 31 datasets with all possible combinations of five subjects and employed SVM on them. The performances of models trained on different numbers of subjects were compared. For all models in this study, the *R*2 score was chosen to be the evaluation metric. The equation of the *R*2 score is as follows:(9)$$R2=1-\frac{\sum_{i=1}^{N}{\left(\hat{y_{i}}-y_{i}\right)}^{2}}{\sum_{i=1}^{N}{\left(\bar{y}-y_{i}\right)}^{2}}$$
where *N* is the number of samples and $\bar{y}$ is the mean value of all samples’ real label. For the *i*th sample, $y_{i}$ is the sample’s real label and $\hat{y_{i}}$ is the sample’s predicted value given by a model. The possible value range of the *R*2 score is $\left(-\infty ,\right.\left.1\right]$ and the value reflects the percentage of the change in the dependent variable caused by independent variables to the total change in the dependent variable. In other words, the *R*2 score is the explanatory degree of the independent variable to the dependent variable. If the *R*2 score is 1, we can explain all the changes in the dependent variable with the independent variable and the model can give a predicting result without any error. The closer *R*2 score is to 1, the higher model’s goodness of fit is, which indicates that the model is better. If the *R*2 score is 0, the model shows no effectiveness.

## 3. Results and Discussion

### 3.1. Regression Result

We utilized a series of datasets as an example to verify the feasibility of quantitatively recognizing primary taste stimuli intensity with sEMG. For each primary taste type, the corresponding dataset was comprised of all samples from subject 1. For each taste type and taste intensity, the sample amount of the dataset is shown in Table 3.

It was important to have a label that could tell the model the real value of each sample’s taste intensity. We tried three types of labels, including strength labels, relative concentrations, and scale scores. The strength labels and relative concentrations denoted the concentrations of the solutions. For the solution with the highest concentration, the strength label was set to 5 and the relative concentration was set to 1. When the tastant concentration decreased by half an order of magnitude, the strength label decreased by 1. Meanwhile, the relative concentration was the ratio of tastant concentration to the highest concentration. For instance, the highest concentration of citric acid used in the experiment was 0.2 mol/L. For a sample under stimuli of 0.2 mol/L citric acid (aq), the strength label was set to 5 and relative concentration was set to 1. For a sample under stimuli of 0.02 mol/L citric acid (aq), the strength label was set to 3 and the relative concentration was set to 0.1. In a word, strength labels decreased exponentially with the decrease of the solution’s concentration, and relative concentrations decreased linearly with the decrease of the solution’s concentration. For a sample under stimuli of deionized water, the strength label and relative concentration were both set at 0. The scale score is the score given by the subject during the ninth step in the experiment. SVM was performed on each dataset, giving five-fold cross-validation *R*2 scores shown in Table 4 with each type of label.

For objective descriptions of tastant concentrations, including the strength label and relative concentration, the result showed that the former outperformed the latter, which means that the strength label is a better choice for our recognition tasks. The reason for this result might be that the subjects’ taste sensations declined linearly with the tastant concentration declining exponentially. When we were designing the experiment protocol, we carried out preliminary experiments to decide the decline categories of the tastant concentration. We asked volunteers to taste the tastants double-blindly and give taste intensity scores. If the concentration declined linearly, the volunteers had difficulty telling different tastants apart, while if the concentration declined exponentially, the volunteers could give roughly linear declining scores. The regression result was consistent with our judgment when designing the experiment. Furthermore, the strength label outperformed the scale score. As the subjects reported, it was easy to tell the existence of the differences between tastants, but it was difficult to quantifiably and accurately describe the intensity with scores. For example, a subject could confirm that the intensity of tastant A was between tastant B with a score of 10 and tastant C with a score of 4, but the subject could not choose a clear enough number among 8, 7, or 6 to evaluate the taste intensity of A; as such, the result could be influenced by the subject’s subjective thoughts. Based on the comparisons, we chose the strength label as the regression target value in the following exploration in this study.

The regression results are shown in Figure 5. For each subgraph, the width of the shape is the density of data points. The abscissa coordinate is the true strength label, and the vertical coordinate is the predicted strength label. The taste providing negative hedonic values (Sour, Bitter, and Salty) showed an apparently higher score than that providing positive hedonic values (Sweet and Umami), which is reasonable, given that under stimuli of the former, the subjects made more facial expressions than the latter. The result showed that under the stimuli of Sour, Bitter, or Salty, there is a certain correlation between sEMG patterns and taste intensity, and it is feasible to utilize sEMG to quantitatively recognize taste stimuli intensity. Meanwhile, for Umami, the correlation is not apparent and for that of Sweet, there is almost no correlation. The result of Umami was not so bad, but we considered it not feasible to recognize the taste. The detailed reason for this is given in Section 3.4. As for Sweet, it was infeasible for all subjects. Moreover, the target of this study was to explore whether the sEMG has the potential to be utilized for recognizing the intensity of primary taste stimuli; therefore, we focused on the regression task rather than the classification of different primary tastes.

### 3.2. Facial Expression Differences

To compare the subjects’ facial expressions under different taste intensities, we utilized the spectrum to infer muscle activities. The integral of spectrum amplitude reflects the energy of the sEMG and the energy reflects the muscle activity strength. Therefore, for each channel of each sample, the integral of spectrum amplitude was calculated and marked as *S_t,i,n,c_*, where *t* and *i* are the taste type and taste intensity of the sample, *n* is the index of the sample among samples of *t*th taste type and *i*th taste intensity, and *c* is the index of the channel. For each taste type *T*, taste intensity *I*, and channel *C*, the mean value of channel spectrum amplitude integral was calculated as follows:(10)$$S_{T,I,C}=\frac{\frac{1}{N_{T,I}}\sum_{n=1}^{N_{T,I}}S_{t,i,n,c}\;\left(if\;t=T,\;\;\;i=I\;and\;c=C\right)}{\frac{1}{N_{T,0}}\sum_{n=1}^{N_{T,5}}S_{t,i,n,c}\;\left(if\;t=T,\;\;\;i=0\;and\;c=C\right)}$$
where *N_T,I_* is the sample amount of *T*th taste type and *I*th taste intensity. Respectively, *S_T,I,2_*, *S_T,I,3_*, *S_T,I,4_*, *S_T,I,5_*, and *S_T,I,6_* could reflect the activities of the depressor anguli oris, the levator labii superioris, the risorius, the procerus, and the masseter under stimuli of *T*th taste type and *I*th taste intensity. Both electrodes for channel 1 and channel 6 were arranged on masseters. A signal without differential operation could reflect signal strength more intuitively; therefore, we utilized the integral of spectrum amplitude of channel 6 to infer the activity of the masseter.

After calculating all 150 S values for 5 taste types, 6 taste intensities, and 5 channels, we linearly mapped different S values to different colors based on a color bar. Then, we represented muscle activities in color, as shown in Figure 6 with the color bar. The activity of a muscle is stronger if the color of the muscle is darker in the figure. The figure indicates that for taste types providing negative hedonic values, the depressor anguli oris showed the strongest activity, which presented as an apparent tremor of the underlip and the mouth corner, as observed during the experiment. The muscle activity became stronger when the taste intensity got higher, which means the tremor was more intense. Meanwhile, the activity of the procerus was stronger than the other three types of muscles, and this activity could be manifested by frowning, also observed during the experiment. Similarly, the muscle activity grew stronger when the taste intensity got higher, which means the frown was larger. The activity of the levator labii superioris was related to the quiver of the upper lip, which was also apparently influenced by taste intensity. Lastly, the activities of the masseter and the risorius were not as apparent as other muscles and there were not many observational records of lateral movements of the corner of the mouth or quivering of the cheeks. For all three types of taste types providing negative hedonic values, the patterns of facial expression changing are similar, which could be a general reaction to unpleasant taste stimuli. Muscle activities under the stimuli of different taste intensities of Sweet and Umami did not show apparent differences, which was consistent with the results in Section 3.1.

### 3.3. Feature Dimensionality Reduction

Since sEMG contains redundant information not beneficial for the model, we explored the reduction of the feature dimensionality. Taking the regression task of Sour as an example, we reduced the feature dimensionality from 330 to 210 and increased the *R*2 score from 0.7277 to 0.7582. For each channel, we obtained 55 features. We divided these 55 features into 11 groups in order (five features per group). For instance, group 1 contained features 1–5 and group 2 contained features 6–10. We constructed an empty feature set and added the feature groups one by one into the new feature set. When choosing a feature group, we selected the one that could give the best model performance combed with feature groups that had been added into the new feature set. Taking channel 1 for instance, we performed SVM on each group of features, and group 2 gave the best *R*2 score (0.2267); therefore, group 1 was added into the new feature set. Then, for each one among the other ten groups, we combined it with feature group 2 and performed SVM on it. Feature group 10 gave the best *R*2 score (0.2340); therefore, group 9 was added into the new feature set. We repeated the steps until all 11 feature groups were added to the new feature set. When we added a feature into the new feature set, the *R*2 score increased because we introduced new information for the model. However, when too many features were added to the new feature set, the *R*2 score started to decrease, resulting from the impact of redundant information on the model. We marked the number of features in the new feature set as NF and the *R*2 score given by the model performed on the feature set with NF features was marked as *R*2_NF_. Then, we were able to obtain a curve of the relation between *R*2_NF_ and NF, as shown in Figure 7.

For each channel, among all points satisfying the condition that the difference between the point’s *R*2_NF_ value and the maximum of all *R*2 scores was not larger than 0.02, the one with the largest NF value was chosen as the curve’s inflection point. Taking channel 1 as an example, the maximum value of *R*2 scores was 0.2642 and *R*2_25_ was 0.2591; the difference value is smaller than 0.02. When there were 25 features in the feature set, the model would get worse if we added more features to the feature set. Therefore, the point where NF equaled 25 was chosen as the curve’s inflection point. The same protocol worked on the other five channels, and we were able to get a new feature set by combing all the selected features from all channels together. The reserved features for each channel are listed in Table 5.

Similarly, the relation between *R*2_NF_ and NF for the other four taste types is shown in Figure 8. Then, we performed SVM on new datasets constructed based on the new feature set. For all five tastes, the original *R*2 score, the feature dimensionality after reduction for each channel, the total feature dimensionality after reduction, and the *R*2 score after feature dimensionality reduction are listed in Table 6. For the readability of the table, the feature dimensionality is written as FD for short and the feature dimensionality reduction is written as FDR for short.

The regression results after feature dimensionality reduction are shown in Figure 9. The black shape represents the distribution of data points before feature dimensionality reduction and the red shape represents the distribution of data points after feature dimensionality reduction. For each subgraph, the abscissa coordinate is the true strength label, the vertical coordinate is the predicted strength label, and the width of the shape represents the density of data points. It is evident that the distributions of data points are almost the same before and after feature dimensionality reduction, which means the process of feature dimensionality reduction did not really impact the model’s performance.

The exploration of feature selection is still rough. Firstly, there are many feature types suitable for sEMG analyses which we did not attempt [44]. Secondly, although the feature dimension reduction achieved the expected goal, reducing the feature dimension without affecting the model’s performance, the feature combinations after reduction were different for different channels or different tastes and showed no apparent pattern. The mechanism behind it still needs further study.

### 3.4. Model Performance on Different Subjects

After the analyses conducted in Section 3.2 and Section 3.3, we got a smaller feature set for each taste type’s model. Then, we tested the model performance on other subjects or multiple subjects. Similar to Section 3.1, we constructed a dataset series for each other subject, and the sample amounts for these datasets are shown in Table 7.

To explore the model’s performances on other subjects, we performed SVM on the datasets of all five subjects. The result for five taste types and five subjects are listed in Table 8 and relatively small values are marked in red. It can be stated that the regression results were influenced by the subject’s character, in that different subjects have different facial expression responses to taste stimuli. Therefore, the model will require preliminary training before actually being applied. For Umami, only subject 1 had an expected result. As we recorded during the experiment, all other subjects reported that the taste Umami provides a neural or positive hedonic value, but subject 1 disliked the taste Umami and reported a strongly negative hedonic value. Therefore, subject 1 showed a more promising result for Umami. The result is consistent with the conclusion in Section 3.1 that the model is feasible for tastes providing negative hedonic values. This is the reason why we considered it is not feasible to recognize the taste of Umami in Section 3.1. This case indicates again that the model’s performance was influenced by each subject’s unique personality and required preliminary training.

Then, we explored the relationship between the model performance and subject diversity by looking at the number of subjects involved in the model. If we picked one subject from five subjects, there would be five possibilities. If we picked two subjects from five subjects, there would be 10 possible combinations. Similarly, there would be 10 possible combinations for three subjects and five possible combinations for four subjects. Therefore, for all five subjects, there would be 31 possible combinations (including the combination of all subjects). We constructed a dataset series for each combination among them. For a combination of more than two subjects, the dataset for the combination was a datasets mixture of subjects in the combination. For each subject combination, the number of subjects was marked as NS (NS = 1, 2, … 5). Then, we performed SVM on these 31 dataset series and obtained *R*2 scores. For all combinations of NS subjects, the mean value of *R*2 scores was marked as *R*2_NS_. Taking NS equaling 2 and Sour as an example, we constructed dataset series of Sour for all 10 possible combinations of two subjects among five subjects. Then, we performed SVM on these 10 datasets and obtained 10 *R*2 scores. The mean value of these 10 *R*2 scores was marked as *R*2_2_ for Sour. It was the same for the other NS values and tastes. We determined the relation between NS and *R*2_NS_ for each taste, which is shown in Figure 10.

It can be stated that for the NS, the results of Sour and Bitter were the best, followed closely by that of Salty, and the results of Umami and Sweet were not satisfying enough. For all taste types, the *R*2 score decreased with the increment of NS. This indicates that for our limited subjects, the model performance decreased with the increment of subject diversity in the dataset, which was consistent with a general law of pattern recognition. The stable level of the *R*2 score could not be confirmed with our limited subjects. The limitations call for further exploration in future research.

## 4. Conclusions

In this study, the feasibility of taste stimuli intensity quantitative recognition based on sEMG was explored. Based on liquid tastants and filter paper, we designed an experimental protocol to acquire sEMG under stimuli of different taste intensities. After the data acquisition, the signals were augmented to samples. Then, the samples’ baseline wandering was removed with a QVR method and power frequency interference was removed with an adaptive notch. After preprocessing, the samples were regressed with an SVM regressor, which gave five-fold cross-validation *R*2 scores of 0.7277, 0.1963, 0.7450, 0.7642, and 0.5055 for the taste types of Sour, Sweet, Bitter, Salty, and Umami, respectively. The feasibility of quantitative recognition of Sour, Bitter, and Salty was verified. As far as we know, this is the first study investigating the quantitative recognition of taste stimuli based on sEMG. Compared to other studies, such as those by Hu and Horio [37,38], which analyzed the sEMG responding to taste stimuli, this study went one step forward, recognizing the taste stimuli based on sEMG. Compared to taste recognition based on sensors, such as artificial tongues, biomimetic sensors, and biosensors [2,7], this study utilized physiological signals other than sensor response; therefore, the system proposed in this study could be applied to many types of tastants or stimuli resource without tastants. Although this study only proposed the recognition of taste stimuli intensity with the same primary taste type, it could provide a foundation for recognizing complex taste stimuli in the future.

To further explore related muscle activities, the spectrum amplitude integral value was calculated and mapped to a color bar for visualization. The activities of five related muscles were summarized and the result was consistent with the results of the previous section. The main difference in muscle activity was that of the depressor anguli oris, which performed as a tremor of the underlip and the mouth corner. The activity of procerus was also intense, manifesting as a frown. Related to the quiver of the upper lip, the activity of levator labii superioris also showed some differences. All of these results were consistent with the observations in the experiment.

To simplify the model, for each taste type and each channel, we constructed a new dataset of features from this channel and removed them one by one to find an optimized number and combination of features. Then, for each taste type, we combined new features of each channel together to get a new feature set. We reduced the feature dimensionalities from 330 to 210, 120, 210, 260, 170 for Sour, Sweet, Bitter, Salty, and Umami, respectively.

Lastly, we tested the model performance on datasets with samples from different numbers of subjects. Judging from the limited subjects we had so far, there was a decrement in model performance when more subjects were introduced to the dataset.

The main research goal of this study was to propose a novel method for the quantitative recognition of primary taste stimuli intensity. The study proved the feasibility of the recognition tasks of the primary taste types of Sour, Bitter, and Salty, which could provide a potential approach to quantitatively describe the stimuli intensity of these three types of primary tastes.

However, there are still many limitations in our study. Firstly, the recognition results of taste types Sweet and Umami were not satisfying enough, which could be due to the lack of facial expression under the stimuli of taste providing positive hedonic values. Meanwhile, the results of Sour, Bitter, and Salty were influenced by each subject’s personality and personal preference. This shortcoming could be solved by the abundant information of EEG. We will explore the possibility to combine the sEMG and EEG to find a balance between the convenience of sEMG and the information richness of EEG. Secondly, there was a lack of an investigation of the electrode sizes, positions, and inter-distances, which could influence the signals and the recognition results [45,46]. Thirdly, different parts of the tongue have different susceptibility to basic taste stimuli, and the effect of stimuli position should be tested. Fourthly, the study lacked an exploration of feature set selection. The feature set of this study was simple and there are more reliable feature sets that might improve the model’s performance [44]. Fifthly, there are still some aspects that could be further optimized during the preprocess, such as the window size and step size of the sample augment. Lastly, the number of subjects and samples is not enough, and it is necessary to explore the effect of subject diversity on model performance.

## Figures and Tables

**Figure 1 sensors-21-06965-f001:**
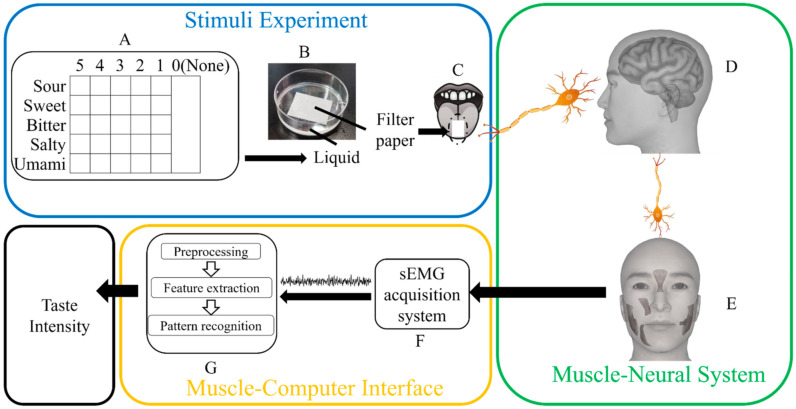
Schematic diagram of the study. Twenty-six types of solutions (including deionized water) were prepared (**A**). The filter paper was soaked in a type of solution for each trial (**B**). Then, the filter paper was put on the subject’s tongue (**C**). The signal of taste sensation was transported to the central neural system (**D**). Driven by the central neural system, the muscles were activated (**E**). An acquisition system recorded the sEMG of these muscles (**F**). The taste intensities were quantified based on acquired signals, after signal preprocessing, feature extraction, and pattern recognition (**G**).

**Figure 2 sensors-21-06965-f002:**
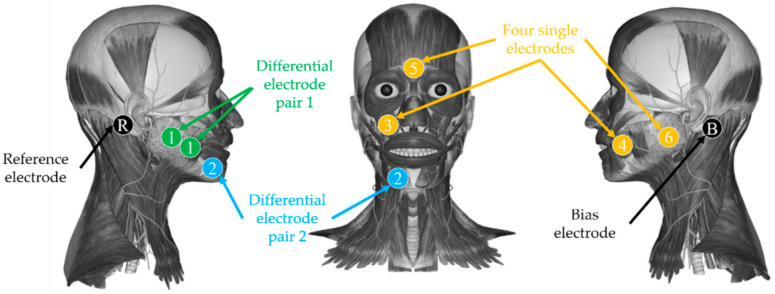
Electrode distribution. The circles indicate the position of electrodes, and the numbers denote the index of 6 channels. Electrodes for channels 1 and 2 are differential electrode pairs. Circles with the letter ‘B’ and ‘R’ are the positions of the bias electrode and reference electrode. Electrodes for channels 1 and 6 were arranged on masseters. Electrodes for channel 2 were arranged on depressor anguli oris. Electrodes for channels 3, 4, and 5 were arranged on levator labii superioris, risorius, and procerus, respectively. In addition, the bias electrode and reference electrode were arranged on the left and right mastoid, respectively.

**Figure 3 sensors-21-06965-f003:**
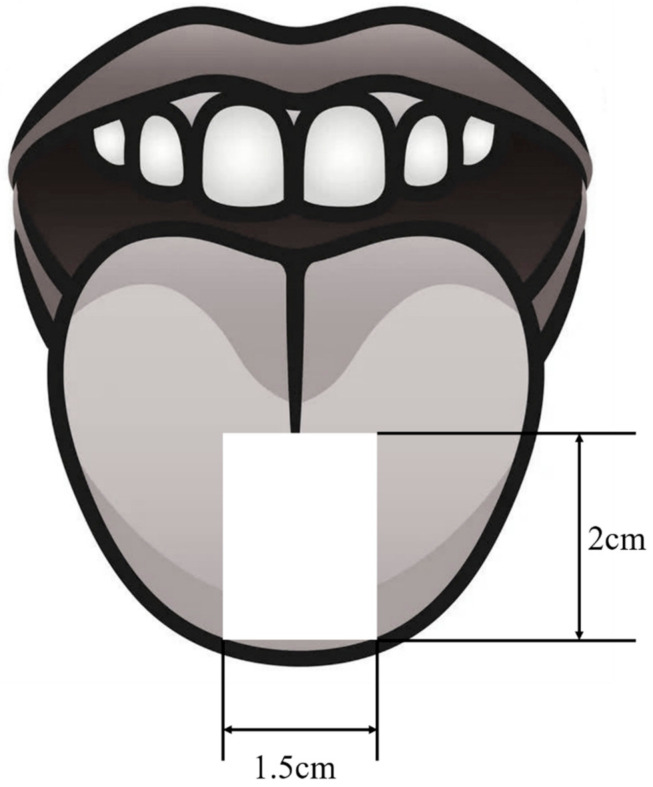
Stimuli position.

**Figure 4 sensors-21-06965-f004:**
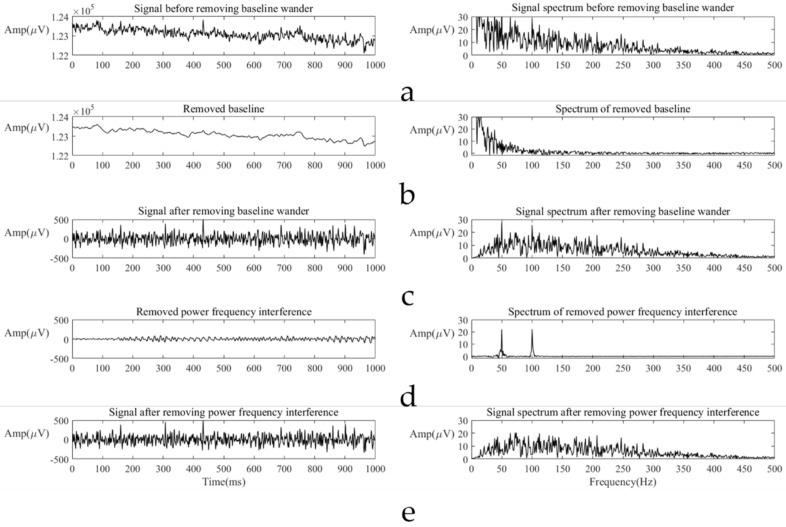
An example of removing baseline wandering and power frequency interference. Preprocessing result of a signal includes the signal before preprocessing (**a**), removed baseline (**b**), the signal after removing baseline (**c**), removed power frequency interference (**d**), and the signal after preprocessing (**e**). For each subgraph, the left graph is the time domain signal, and the right graph is the spectrum of the signal. In the spectrum of subgraph (**b**,**c**), the upper limit on the y axis was fixed 30 μV, because the amplitude in this region is too large (more than 100 mV) and contains no information but baseline wandering.

**Figure 5 sensors-21-06965-f005:**
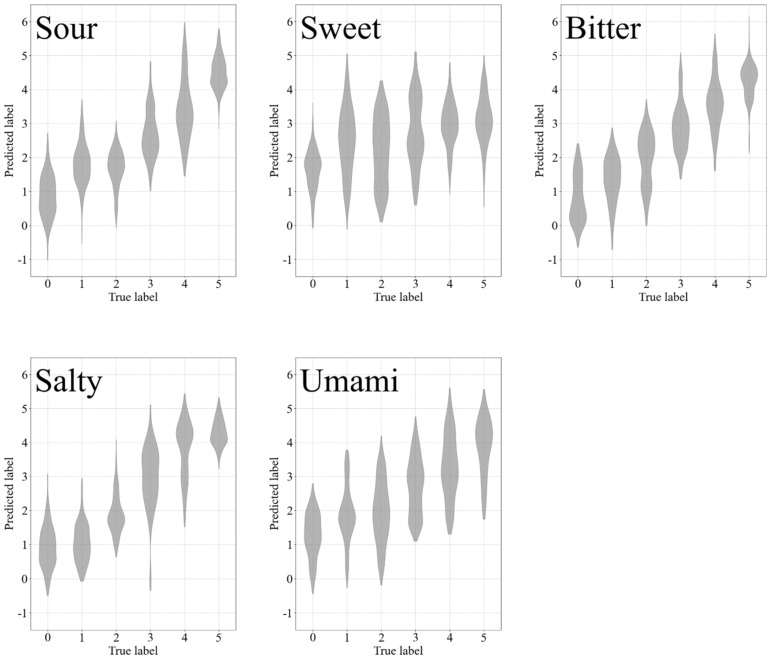
Regression results. The regression result for each primary taste type is shown with a violin plot and the width of the shape is the density of data points. The abscissa coordinate is the true strength label, and the vertical coordinate is the predicted strength label.

**Figure 6 sensors-21-06965-f006:**
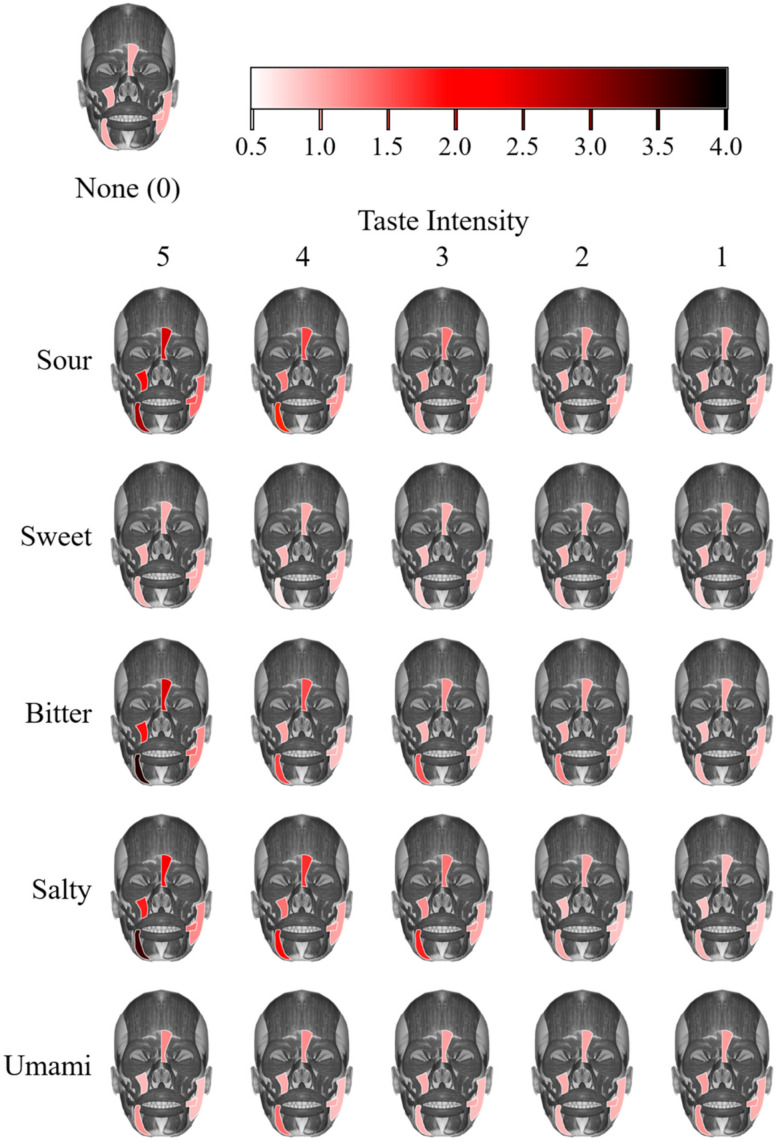
Muscle activity under stimuli. The muscle activities were reflected with the mean integral values of spectrum amplitude under different taste stimuli. The activities of the depressor anguli oris, the levator labii superioris, the risorius, the procerus, and the masseter were calculated based on channels 2, 3, 4, 5, and 6, respectively. For each muscle, the integral value was normalized by dividing by the value under stimuli of None (taste intensity 0) and mapped to colors based on the color bar shown at the top of the figure. A darker color of a muscle indicates a stronger activity of the muscle. The muscle activity under stimuli of None (all equaled 1 after normalization) is shown at the upper left corner of the figure.

**Figure 7 sensors-21-06965-f007:**
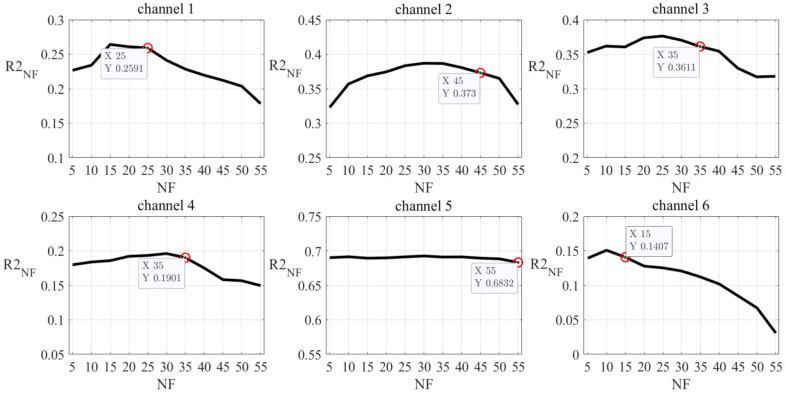
Relation between *R*2_NF_ and NF for taste Sour. For each channel, datasets only containing features from this channel were constructed. NF is the number of features and *R*2_NF_ is the *R*2 score of SVM performed on a dataset containing NF features. The red circles are the curves’ chosen inflection points. The figure shows that the trend of *R*2_NF_ changed as the NF changed.

**Figure 8 sensors-21-06965-f008:**
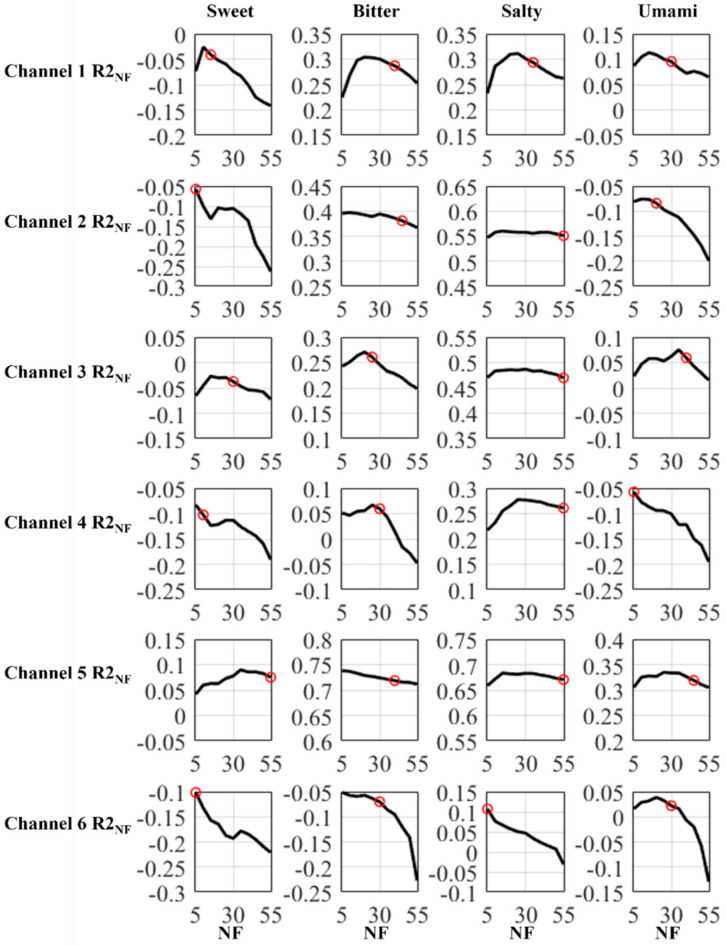
Relation between *R*2_NF_ and NF for Sweet, Bitter, Salty, and Umami. For each taste type and each channel, datasets only containing features from this channel were constructed. NF is the number of features and *R*2_NF_ is the *R*2 score of SVM performed on a dataset containing NF features. The figure shows the trend of *R*2_NF_ changed as the NF changed. The red circles are the curves’ chosen inflection points.

**Figure 9 sensors-21-06965-f009:**
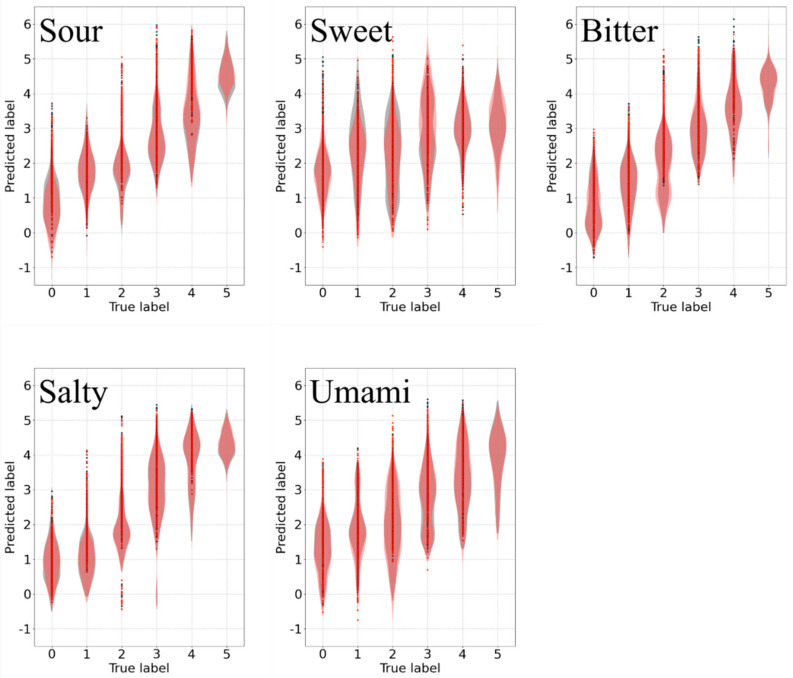
Regression results after feature dimensionality reduction. The regression result for each primary taste type is shown with a violin plot and the width of the shape is the density of data points. The black shape represents data points before feature dimensionality reduction and the red shape represents data points after feature dimensionality reduction. The abscissa coordinate is the true strength label, the vertical coordinate is the predicted strength label.

**Figure 10 sensors-21-06965-f010:**
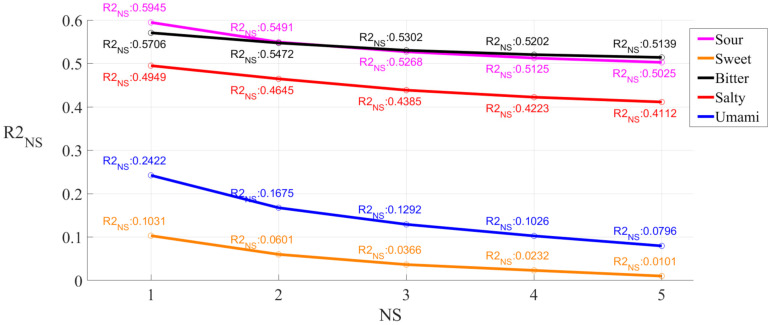
Relation between *R*2_NS_ and NS. NS is the number of subjects. We listed all possible subject combinations of NS subjects for each NS (NS = 1, 2, … 5) and constructed a new dataset series for each combination by combining datasets series of subjects in the combination. After performing the SVM, we could get an *R*2 score for each combination and each taste type. For each type and each NS, the mean value of the *R*2 scores was marked as *R*2_NS_. The figure shows that the trend of *R*2_NS_ changed as the NS changed.

**Table 1 sensors-21-06965-t001:** Detailed electrode positions.

C	Electrode Type	Muscle	Detailed Position *
1	Differential electrode pair	Right masseter	From the lower end of the right ear cartilage, 3 cm forward.
2	Differential electrode pair	Right depressor anguli oris	Below the right corner of the mouth
3	Single electrode	Right levator labii superioris	1 cm over the upper lip and tangent to the right alar of the nose.
4	Single electrode	Left risorius	From the left corner of the mouth, 1 cm to the left
5	Single electrode	Procerus	From the midpoint of the line between the two inner ends of the eyebrows, 2 cm upward.
6	Single electrode	Left masseter	From the lower end of the left ear cartilage, 3 cm forward.
B	Bias electrode	Left mastoid	The protuberance behind the left ear
R	Reference electrode	Right mastoid	The protuberance behind the right ear

* For a pair of differential electrodes, the detailed position referred to the tangent point of two circular electrodes’ adhesive rings, while for a single electrode, the detailed position referred to the center of the circular electrode.

**Table 2 sensors-21-06965-t002:** The tastants providing different taste types and intensities.

Taste Type	Tastant (aq *)	Concentration (mol/L) for Different Strength Label					
		5	4	3	2	1	0
Sour	Citric acid	0.200	0.063	0.020	0.006	0.002	0
Sweet	Sucrose	1.000	0.316	0.100	0.032	0.010	0
Bitter	Magnesium chloride	1.000	0.316	0.100	0.032	0.010	0
Salty	Sodium chloride	2.000	0.632	0.200	0.063	0.020	0
Umami	Sodium glutamate	1.000	0.316	0.100	0.032	0.010	0

* The solvent of all solutions was deionized water and a solution of 0 mol/L was just deionized water.

**Table 3 sensors-21-06965-t003:** Number of samples in datasets for different taste types.

Strength Label	Relative Concentration	Sour	Sweet	Bitter	Salty	Umami
5	1	479	455	468	465	451
4	0.316	452	474	450	475	460
3	0.1	459	462	474	453	466
2	0.0316	469	478	466	481	454
1	0.01	474	477	471	458	452
0	0	475	454	467	467	467
All		2808	2800	2796	2799	2750

**Table 4 sensors-21-06965-t004:** Regression result (*R*2 scores) for different taste types and label types.

Label Type	Sour	Sweet	Bitter	Salty	Umami
Strength label	0.7277	0.1963	0.7450	0.7642	0.5055
Relative concentration	0.7050	-0.0590	0.6039	0.5000	0.2446
Scale score	0.6506	0.2164	0.6986	0.6950	0.2721

**Table 5 sensors-21-06965-t005:** Reserved features during the feature dimensionality reduction for Sour.

Channel Index	Reserved Feature Index	Reserved Features
1	1–15 36–40 46–50	The spectrum from 0 to 149 Hz; The spectrum from 350 to 399 Hz; The spectrum from 450 to 499 Hz
2	1–40 46–50	The spectrum from 0 to 399 Hz; The spectrum from 450 to 499 Hz
3	11–15 21–50	The spectrum from 100 to 149 Hz; The spectrum from 200 to 499 Hz
4	11–20 26–50	The spectrum from 100 to 199 Hz; The spectrum from 250 to 499 Hz
5	1–50 51–55	The spectrum from 0 to 499 Hz FC, RMSF, RVF, RMS, MAV
6	31–40 51–55	The spectrum from 300 to 399 Hz FC, RMSF, RVF, RMS, MAV

**Table 6 sensors-21-06965-t006:** Results of the feature dimensionality reduction.

Taste Type	*R*2 Score before FDR *	FD * for Each Channel after FDR						Total FD after FDR	*R*2 Score after FDR
		1	2	3	4	5	6		
Sour	0.7277	25	45	35	35	55	15	210	0.7582
Sweet	0.1693	15	5	30	10	55	5	120	0.2272
Bitter	0.7450	40	45	25	30	40	30	210	0.7603
Salty	0.7642	35	55	55	55	55	5	260	0.7793
Umami	0.5055	30	20	40	5	45	30	170	0.5187

* FD is short for the feature dimensionality and FDR is short for the feature dimensionality reduction.

**Table 7 sensors-21-06965-t007:** Sample amounts for all subjects.

Subject Index	Sour	Sweet	Bitter	Salty	Umami
1	2808	2800	2796	2799	2750
2	2801	2806	2837	2800	2863
3	2822	2804	2808	2811	2760
4	2757	2779	2771	2771	2768
5	2758	2779	2748	2769	2773
All	13,946	13,968	13,960	13,950	13,914

**Table 8 sensors-21-06965-t008:** Regression results for all subjects.

Subject Index	Sour	Sweet	Bitter	Salty	Umami
1	0.7554	0.1770	0.7607	0.7729	0.5384
2	0.4960	0.1132	0.6215	0.3062	0.3196
3	0.5405	0.0294	0.5794	0.4819	0.1213
4	0.6931	0.2690	0.4470	0.5031	0.3742
5	0.5032	0.1150	0.4308	0.4605	0.3178

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
