# Peer review of "Quantitatively Recognizing Stimuli Intensity of Primary Taste Based on Surface Electromyography"

_sensors, 2021, doi:10.3390/s21216965_

Round 1
Reviewer 1 Report
1.Different parts of the tongue have different susceptibility to basic taste stimuli, but this article only carried out test on one part of the tongue? In addition, the sample is too small, experiment was carried out only for 5 subjects. Individual differences were difficult to reflect.
2.Why these kinds of seasonings and additives were chosen as primary taste stimulation? Can they be replaced with other additives? Why these concentrations (99.5%, 99.6%, etc.) were chosen to provide primary taste?
3.In the introduction part, there is too much introduction about bci, and too little introduction about taste recognition of electromechanical signals.
4.Lines 71-76 and reference 34-38 refer to the current research status of gustatus-related studies, without mentioning the shortcomings of each study or the advantages of the research algorithm proposed in this paper compared with existing research methods.
5.The subgraphs of Figure 4 should be separately labeled as (a) (b), (c)… instead of being mixed together and labeled with row3, etc.
6.Lines 294-295 mentioned “taste the tastants double-blindly and give the taste intensity scores”. It is not included in the experimental procedure. Please complete the experimental procedure.
7.It is mentioned in line 382 that “ the taste intensity threshold where the facial expressions start to appear could be different”. But there is no introduction about this part in other parts of the text. It is recommended to supplement the threshold setting.
8.Lines 389-390 say that “only some of the subjects recognize the taste of umami” Actually only experimenter 1 recognized umami, so dose it means that it is very likely that umami cannot be recognized?
9.What is the meaning of 31 combination of subjects mentioned in line 395?
10.In addition, primary taste correction is not involved, so this paper should propose a more advanced method to objectively define primary taste
Author Response
We really appreciate your careful work and kind guidance, and it helps us a lot. Please check the attachment to find point-to-point responses.

Reviewer 2 Report
The paper describes an expert system that correlates facial expressions evoked by taste stimuli to a quantifier of the substance used in the stimuli using EMG features of the facial muscles. The stimulus methodology is well designed and the topic may be of interest for physiology and behaviors researchers. The paper lacks in the EMG processing techniques, as is far from the state-of-the-art methods. However, there is some novelty as the quantification of human behavior is somehow a "hot topic" in the field. Moreover, the graphical aspect of the paper has its merit. Nevertheless, some issues must be addressed:
Major issues:
- The introduction has a lot of information but there is space for improvement:
* Expand the literature review about what methods have been used for the identification of taste stimuli.
* As the paper describes a facial expression pattern recognition system using EMG, it is a good idea to present some relevant papers about it. There are some in the literature but none is cited;
* The topic is closed related to chewing analysis using EMG, there is a lot of works about the physiology and behavior of mastication (human and animals). For example, several works correlate food properties to EMG features which could be used or cited by the authors;
* The paragraph about the BCI application is huge! Reorganize the ideas you want to present in a way each paragraph tells a concise and precise piece of information. Avoid huge paragraphs with spitted citations. Elaborate what the cited authors did, it is important to compare to your work later in the discussion;
- It is not a bad idea to present the contributions at the end of the introduction, but details about the results are rarely in the introduction. At the end of the introduction, you may present the problem and how this work is solving the gap. Results must be in the results section (and briefly described in the abstract).
- The pre-processing is odd. The authors overjustify the use of a powerline notch filter (which necessity is exhaustively known in the literature). However, there is no consideration about the decision on the window size/step size. Moreover, why did the authors decided to use the QVR? Usually, the procedure is to use a Butterworth band-pass filter (usually from 20-450 Hz). I think it is not a big deal, but it does need some justification.
- Does the electrode placement follows a standard procedure (such as SENIAM http://www.seniam.org/)?
- What were the criteria for choosing the features? There are several well-known feature sets for EMG that must be considered, e.g.:
https://www.mdpi.com/1424-8220/18/5/1615/pdf
-Moreover, have the authors considered testing features used in facial expression recognition using EMG? If the intention was to propose a novel feature set for the purpose, a feature selection is needed to reduce the dimensionality of the input variables.
- Is there discrimination between the classes (or there is a study about it)? Before correlating a quantifier, it may be useful to known if there is a class separability for the primary tastes. This can be achieved by the performance of a classifier or from a PCA. If the intention is to correlate the quantifier, regardless of the type of taste, this may not be necessary, but it should be explained in the text.
Minor issues:
- Check the formatting of the references for the journal (there is a space between the numbering and the text).
- Units must be separate from values, e.g. 1000 Hz.
- In Figure 2 illustrate which electrodes are differential.
Author Response

(The authors gave the same response as above.)

Reviewer 3 Report
The paper presents a method for recognizing taste stimuli by application of surface electromyography recordings of facial muscles. Interesting visualization approach and Support Vector Machine were applied. However, paper needs improvements both major and minor.
Major comments:
- My main concern is related to the taste intensity. It is highly subjective as it depends on saliva, see for example: Matsuo, R. "Role of saliva in the maintenance of taste sensitivity." Critical Reviews in Oral Biology & Medicine 11.2 (2000): 216-229. How did authors control saliva influence in their design?
- Is the detailed position of electrodes given in Table 1 in accordance with the SENIAM protocol? If not, what is its relation to SENIAM and why did you use locations presented in Table 1? If yes, please add this information in the manuscript.
- How did authors decide on the pause duration during the steps of measurement procedure?
- What is the rationale for 330 features and their relation to the facial expression? Would it be better to apply feature selection method?
- Comparison with existing methods for surface electromyography application for taste classification is missing in the discussion.
Minor comments:
- BCI-related research is irrelevant and excessive in the Introduction. Please, either delete it or shorten it.
- How did Authors synchronize the tasting of filter paper with sEMG recordings?
- In Fig. 1 (E) there is a typo and I would suggest to add muscle-neural system instead of just neural system. Also, calling sEMG based system a BCI in Fig. 1 (G, F) is wrong, please use another term.
- Units are always separated from numerical values. Instead of "1000Hz", please write "1000 Hz" and apply it throughout the paper.
- I guess that Authors didn't mean to say "the subject was forbidden to eat" rather "the subjects were asked not to eat" as the second is in line with the Ethical board approval and the Declaration of Helsinki? The same applies for "forbidden to deliberately control".
- There is no length of 10 Hz, please correct this in line 233
- Methods, Results, and Discussion are mixed in the Results section - please, move the appropriate parts to other sections. For example, lines 284-288 should be moved to Methods. This section is not clear and it is hard to follow.
- What is the meaning of red numbers in Table 6? Please, add information.
- In Conclusion: why not combine electrogastrography (EGG) with sEMG?
Author Response

(The authors gave the same response as above.)

Round 2
Reviewer 1 Report
Although there were limitations of stimulus location and sample size, the comments previously proposed have been solved.
But there are still some small problems that need explaining.
Line 86 mentions that some of the shortcomings of EEG can be improved by SEMG, but it does not specify which shortcomings can be improved.
Lines 69 to 95, the disadvantages of each cited reference were not presented. Only a summarize of deficiencies were given.
Author Response
We really appreciate your careful work. Thanks to your questions, we learned much and modified many mistakes, and that really made our article better. Please check the point-by-point responses in the attachment.

Reviewer 2 Report
All my questions were ansewered and all issues were addressed. In my opinion the paper can be accepted.
Author Response
We really appreciate your careful work. Thanks to your questions, we learned much and modified many mistakes, and that really made our article better.
Reviewer 3 Report
Authors have answered my queries appropriately. I have no further suggestions.
Author Response

(The authors gave the same response as above.)
